# SimSCOOD: Systematic Analysis of Out-of-Distribution Generalization in Fine-tuned Source Code Models

## Abstract

Large code datasets have become increasingly accessible for pre-training source code models. However, for the fine-tuning phase, obtaining representative training data that fully covers the code distribution for specific downstream tasks remains challenging due to the task-specific nature and limited labeling resources. Moreover, fine-tuning pretrained models can result in forgetting previously acquired pre-training knowledge. These lead to out-of-distribution (OOD) generalization issues with unexpected model inference behaviors that have not been systematically studied yet. In this paper, we contribute the first systematic approach that simulates various OOD scenarios along different dimensions of source code data properties and study the fine-tuned model behaviors in such scenarios. We investigate the behaviors of models under different fine-tuning methodologies, including full fine-tuning and Low-Rank Adaptation (LoRA) fine-tuning methods. Our comprehensive analysis, conducted on four state-of-the-art pretrained models and applied to two code generation tasks, exposes multiple failure modes attributed to OOD generalization issues. Additionally, our analysis uncovers that LoRA fine-tuning consistently exhibits significantly better OOD generalization performance than full fine-tuning across various scenarios.

## 1 Introduction

There has been increasing success in applying Large Language Models (LLMs) to various source code understanding and generation tasks. LLMs for codes such as CodeBERT (Feng et al., 2020), GraphCodeBERT (Guo et al., 2021), CodeT5+ (Wang et al., 2023), CodeGen (Nijkamp et al., 2023), and Code Llama (Rozière et al., 2023) are pretrained using large-scale source code datasets, and serve as universal initialization for a variety of downstream tasks. These tasks include code summarization (Alon et al., 2019; LeClair et al., 2020), text-to-code (Iyer et al., 2018), code translation (Nguyen et al., 2013; Rozière et al., 2020), and program repair (Tufano et al., 2018; Chen et al., 2019; Hajipour et al., 2021).

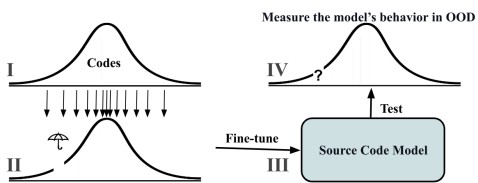

Figure 1: Our approach simulates out-of-distribution (OOD) scenarios and analyzes the corresponding behaviors of models. (I) Original source code distribution along a certain dimension. (II) OOD simulation by masking out a sub-region of the distribution. (III) Model fine-tuning. (IV) Evaluation on OOD data.

The emerging abilities of LLMs, such as in-context learning, demonstrate their potential to handle a wide range of tasks (Wei et al., 2022; Brown et al., 2020). However, it has been shown that not all tasks can be effectively addressed by relying only on the pretrained LLMs (Anil et al., 2022). To adapt pretrained models for specific tasks, they can be fine-tuned with specific datasets for each downstream task. This fine-tuning process can involve optimizing all parameters or adopting a parameter-efficient approach (Houlsby et al., 2019; Hu et al., 2022), such as Low-Rank Adaptation (LoRA)(Hu et al., 2022). Considering fine-tuning is prone to catastrophic forgetting (Chen et al., 2020; Li et al., 2022a), and these models play crucial roles in automatic software development, it

is equally important if not more, to foresee and understand any unexpected models behaviors in different scenarios beyond in-distribution fine-tuning data.

Despite having access to the large code datasets to pre-train these models, it remains challenging in practice to fully cover the code distribution, specifically in fine-tuning datasets, where the availability of labeled data is limited. This mainly stems from the compositional structures of programs and the complexity of software. Furthermore, it has been shown that fine-tuned models forget previously learned knowledge (Chen et al., 2020), and fully fine-tuning the parameters of the pretrained models can distort the pretrained features (Kumar et al., 2022).

Therefore, it is unclear how the fine-tuned code models generalize to scenarios not seen or are rare in the fine-tuning distribution (Shen et al., 2021). For example, there is a lack of existing studies to uncover how these models generalize to programs with specific language elements or semantics not seen in fine-tuning datasets. A common way to study model behaviors in various OOD scenarios is to collect testing datasets in the complementary domains of the fine-tuning dataset domain (Shen et al., 2021). However, because the underlying true distribution of source code is intractable, it is barely feasible to justify whether two raw datasets share a domain or not. Not to mention the substantial costs to enumerate and constitute a variety of OOD testing datasets.

Simulating various OOD scenarios by masking out sub-regions of training data distribution is an alternative way to systematically study the model behaviors (Schott et al., 2022; Wiles et al., 2022). There are several distribution dimensions based on data properties. In the source code domain, we have access to the structural information to model the source code distribution based on the **length**, **syntax**, and **semantics** of programs. For example, in terms of the syntax dimension, we can mask out all the data with *uniray expressions* or specific API to create a syntax-based OOD scenario.

In this work, we propose a systematic approach to analyzing the behaviors of fine-tuned source code models in various OOD and few-data regime scenarios. We achieve this by harnessing the token size, syntax information, and contextual embeddings of programs to simulate the OOD scenarios in terms of length, syntax, and semantics dimensions, as illustrated in Figure 1. By utilizing these data dimensions and control over the data, we can systematically examine the performance of fine-tuned models in OOD scenarios and investigate the generalization capabilities of different fine-tuning methods.

To summarize, the main contributions of this paper are as follows: 1. Our work pioneers in investigating the behaviors of the fine-tuned source code models in OOD scenarios. 2. We propose a systematic approach to simulate various OOD scenarios by masking out sub-regions of source code distribution along the length, syntax, and semantics dimensions. At the time of publication, we will publish the implementation of our work. 3. We find that the performance of the fine-tuned models can significantly deteriorate in various OOD scenarios despite the model encountering similar examples during the pre-training phase. In particular, in syntax and length-based OOD scenarios, the drop can be as substantial as 90%. 4. Our systematic analysis shows that, while full fine-tuning and LoRA fine-tuning perform comparably on in-distribution code data, LoRA fine-tuning demonstrates significantly better performance on OOD data. 5. Our analysis of data/model properties provides insights into model finetuning and shapes future datasets/research to focus on OOD of code models, which has the potential to enhance generalization accuracy across various code generation tasks.

## 2 RELATED WORK

**Large Language Models for Codes.** With the availability of large-scale code datasets (Husain et al., 2019; Kocetkov et al., 2022), there is growing interest in employing large language models to develop a unified pre-training model for source code understanding and generation. Code-BERT (Feng et al., 2020) is one of the first models that use pre-training in the source code domain. CodeBERT extends the RoBERTa-based model (Liu et al., 2019) to understand and generate source code in various programming languages. Guo et al. (2021) extend CodeBERT by using a semantic-aware objective function. CodeT5 and CodeT5+ (Wang et al., 2021; 2023) are developed based on encoder-decoder architecture, making them versatile models for addressing a wide range of code understanding and code generation tasks. Svyatkovskiy et al. (2020) employ GPT-based (Radford et al., 2019), which uses decoder-only architecture, for the code completion task. CodeGen (Nijkamp et al., 2023), StarCoder (Li et al., 2023), and Code Llama (Rozière et al., 2023) also employ

decoder-only architecture to pre-train code generation models, these models demonstrate impressive results across a variety of code generation tasks. While these models show remarkable results by following natural language instructions, it has been demonstrated that LLMs still have difficulty in understanding the codes (Austin et al., 2021; Li et al., 2022b), specifically in domain-specific tasks (Anil et al., 2022). In our work, we focus on generation tasks to spot weak and strong points of the fine-tuned LLMs in generating rare and unseen programs.

**Out-of-Distribution Analysis in Natural Languages and Programming Languages.** Despite the importance of OOD analysis and detection in production (Shen et al., 2021), there are surprisingly much fewer efforts to investigate OOD behaviors of NLP and PL approaches (Arora et al., 2021). Hendrycks et al. (2020); Kong et al. (2020) study the behavior of pretrained large language models in OOD scenarios. Even though they show that the pretrained models are better calibrated, the provided results indicate that there is still room for improvement. Bui & Yu (2021) propose an energy-bounded-based approach to detect OOD data in source code classification tasks. Their approach defines OOD scenarios by masking out data belonging to specific class(es) (Bui & Yu, 2021). Recently, Shi et al. (2022) proposed a set of pre-defined scenarios to investigate the compositional generalization of neural program synthesizers. It is important to note that their investigation was limited to domain-specific languages, such as SCAN (Lake & Baroni, 2018), and did not incorporate pretrained code models. In contrast, we proposed the first systematic study to investigate the behavior of fine-tuned code models across different OOD scenarios.

**Fine-tuning LLMs and Catastrophic Forgetting.** LLMs have demonstrated impressive capabilities in handling various tasks using zero-shot and few-shot learning approaches (Brown et al., 2020; Kojima et al., 2022). However, not all tasks can be effectively handled by relying on pretrained LLMs (Anil et al., 2022; Scialom et al., 2022). For such tasks, we can employ fine-tuning techniques with the datasets for the targeted downstream tasks. Furthermore, recent works indicate that fine-tuning LLMs with instructions can enhance their capabilities (Ouyang et al., 2022; Xu et al., 2023; Dai et al., 2023). Despite the effectiveness of the fine-tuning procedure, recent work shows that after fine-tuning, the LLMs can experience catastrophic forgetting in various NLP tasks (Luo et al., 2023; Chen et al., 2020). Furthermore, Kumar et al. (2022) validates that fully fine-tuning the models can distort the pretraining feature and adversely impact the OOD generalization performance in image classification tasks. In this work, for the first time, we systematically investigate the behavior of the fine-tuned source code models by carefully designing various OOD scenarios.

## 3  SIMSCOOD: SIMULATION OF SOURCE CODE OUT-OF-DISTRIBUTION SCENARIOS

In this work, we propose a systematic approach to investigate the fine-tuned code model behaviors on OOD data by simulating the OOD scenarios in multiple dimensions. Our simulation strategy allows us to construct measurable OOD scenarios without the additional costs of accessing another dataset. More importantly, by simulating the OOD scenarios, we have control over different properties of OOD scenarios. We achieve this by masking out specific sub-regions of data distribution.

These OOD scenarios span over three data dimensions, including **length**, **syntax**, and **semantics**. These dimensions cover different aspects of the programs. In length-based OOD scenarios where we model the program length based on their token sizes (Meneely et al., 2013), we can study the length-generalization ability of the fine-tuned models. For example, whether the models can produce longer codes with high quality and how well the models can interpolate over distribution gaps. Syntax-based scenarios enable us to study the models by masking out specific language elements. More interestingly, using syntax-based scenarios, we can analyze to what extent each model can generate unseen language elements. Using semantic-based scenarios, we can investigate how the models behave if we mask out the data with specific functionalities (e.g., *getter* functions in Java). Benefiting from these scenarios, we can also implicitly quantify how well the models compose different code language elements to achieve unseen or rare functionality.

**Modeling the Distribution of Source Code.** Here, we experiment with different pretrained models and probe their behaviors in each scenario. We achieve this using our new approach that systematically constructs various scenarios to challenge the OOD performance of each model. As a result,

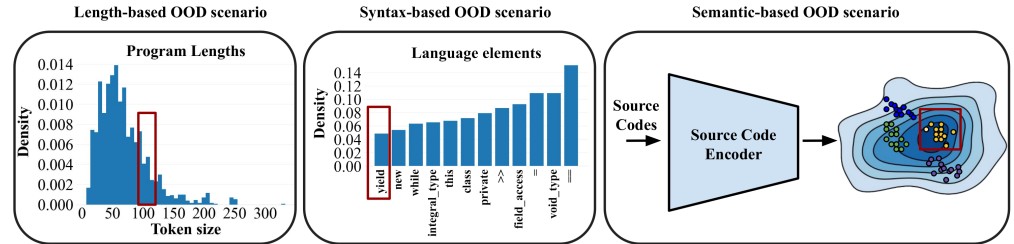

Figure 2: Overview of different out-of-distribution scenarios. Part of the data that needs to be masked out from the training distribution is highlighted by the red rectangles.

the distribution of source code can be characterized using the aforementioned dimensions that we call properties in the following. We model the joint distribution of the source code as $q(p_1, ..., p_n)$ where each $p_i$ is a specific property of the source code in distribution $q$. Given this distribution we can sample a dataset $\mathcal{D} = \{x_1, ..., x_N | x_i \sim q(p_1, ..., p_n)\}$. To create each OOD scenario we need to sample a new dataset $\hat{\mathcal{D}} = \{x_1, ..., x_N | x_i \sim \hat{q}(p_1, ..., p_n)\}$ where $\hat{q}(p_f, ..., p_k) = 0$, meaning the samples with properties $p_f, ..., p_k$ are masked out. Note that we just formulated OOD scenarios with categorical properties, whereas it also holds for continuous properties by $p(a < p_i < b)$ with $a < b$ and $a, b \in \mathbf{R}$.

To sample dataset $\hat{\mathcal{D}}$, we get inspiration from the rejection sampling technique (Casella et al., 2004). Here, $\hat{q}(p_1, ..., p_n)$ is our target distribution and we consider $q(p_1, ..., p_n)$ as our proposal distribution. We reject or accept the sample data $x \sim q(p_1, ..., p_n)$ using the following step function,

$$f(x) = \begin{cases} 1 & \text{if } \mathbf{P}(x) \notin \tilde{\mathcal{P}} \\ 0 & \text{if } \mathbf{P}(x) \in \tilde{\mathcal{P}} \end{cases} \quad (1)$$

Where $\mathbf{P}(x)$ returns the properties of data $x$, and $\tilde{\mathcal{P}}$ are the properties that we do not want the sampled data $x$ to contain. Using the rejection sampling technique with a hard-decision function (Equation 1) we can construct dataset $\hat{\mathcal{D}} = \{x_1, ..., x_N | x \sim \hat{q}(p_1, ..., p_n)\}$ with accepted samples, and also have access to dataset $\tilde{\mathcal{D}} = \{x_1, ..., x_N | x \sim \tilde{q}(p_1, ..., p_n)\}$ which are all of the rejected samples. To examine model behaviors in each OOD scenario, we fine-tune models using $\hat{\mathcal{D}}$ data, and test them on test set of $\tilde{\mathcal{D}}$. Figure 2 depicts an overview of the length-, syntax-, and semantic-based scenarios. In the following, we provide the details of how we simulate each OOD scenario (subsection 4.1).

### 3.1 LENGTH-BASED OOD SCENARIOS

To simulate length-based scenarios, we use the histogram of program token sizes to represent the distribution of a given dataset. See Figure 2 left as an example. To create each OOD scenario, according to the rejection sampling technique, we draw samples from the distribution and reject only the samples in the histogram's specified sub-region.

As an example, in one of the OOD scenarios, we can consider token size between 120 and 135 as OOD testing data. Then $\hat{\mathcal{D}} = \{x \sim \hat{q}(p_1, ..., p_n)\}$ where $\hat{q}(120 < p_i < 135) = 0$ is the accepted data in the rejection sampling technique. Experimenting with the length-based OOD scenarios enables us to analyze how fine-tuned source code models generalize to interpolate and extrapolate over distribution gaps.

### 3.2 SYNTAX-BASED OOD SCENARIOS

Each programming language has its own grammar, which is a set of rules to define valid program statements. Using the grammar, we can parse each program into an abstract syntax tree (Guo et al., 2021) and have access to all of the language elements used in the program. For example, we can identify all the programs with *conditional* or specific APIs in the given dataset. In this work, we leverage the grammatical information of the programming language to create syntax-based OOD scenarios. We use the histogram of language elements to model the syntax distribution of a given

source code dataset. Figure 2 middle shows an example of how we construct a syntax-based OOD scenario by masking out specific language elements. To create an OOD scenario, using the rejection sampling technique, we sample testing data $\tilde{\mathcal{D}}$ that contain certain language elements (e.g., *else*), namely, $\tilde{\mathcal{P}} = \{else\}$. We then fine-tune our model using $\hat{\mathcal{D}}$ which is the set of data that does not contain *else*, and test the model using $\tilde{\mathcal{D}}$. In order to set up systematic syntax-based OOD scenarios, we can replace *else* in $\tilde{\mathcal{P}}$ with other language elements and APIs. Using syntax-based scenarios, in addition to analyzing model behaviors in such OOD scenarios, we can also explore if various fine-tuned LLMs can generate unseen language elements. For example, we can investigate if the pretrained models can generate specific elements and APIs not seen during fine-tuning.

### 3.3 SEMANTIC-BASED OOD SCENARIOS

The programs' semantics is another dimension to model the distribution of source code data. However, it is not clear how we can model the semantics of the programs, especially in the cases where we do not have input-output examples or any meta-data. It has been shown that a large pretrained model can be used to cluster the data based on their semantics (Aharoni & Goldberg, 2020). Furthermore, recent studies conducted by Troshin & Chirkova (2022) and Ahmed et al. (2023) have demonstrated that pretrained code models represent program semantics within the continuous space. They accomplished this by probing the pretrained models and conducting experiments involving the manipulation of code fragments. Following the success of unsupervised domain clustering and the model's abilities to understand the semantics of programs, we propose to utilize the pretrained source code model to cluster programs within the continuous space. We employ the state-of-the-art CodeT5+ encoder (Wang et al., 2023) in our study to map a dataset of programs to a set of continuous representation vectors. We then cluster the vectors to group programs with similar semantics. As a result, we can create semantic-based OOD scenarios via the rejection sampling procedure to reject all samples that belong to a specific cluster and accept the rest as $\hat{\mathcal{D}}$. Like other scenarios, we can use $\hat{\mathcal{D}}$ as fine-tuning data and $\tilde{\mathcal{D}}$ as test data. Our semantic-based OOD scenarios provide an approximated proxy of real-world OOD scenarios to investigate the OOD generalization capabilities of the fine-tuned models. Furthermore, these OOD scenarios allow us to analyze the model's abilities to deal with unseen or rare program functionalities. We provide implementation details in subsection 4.2.

## 4 EXPERIMENTS

In this section, we first articulate the experiment setups, including the pretrained models, downstream tasks, and the OOD data construction process. Then, we demonstrate the model performance in different OOD scenarios. We also analyze how well the model can perform by revealing 50% of the masked data ($\approx 1.5\%$ of the entire fine-tuning data) to the model. In the following, we call the 50% masked-out cases few-data regime.

### 4.1 SETUPS

**Pretrained Models.** We analyze the behavior of four widely-used pretrained models for source codes. These models are designed using a variety of architectures, pre-training objective functions, numbers of parameters, and pre-training datasets. GraphCodeBERT (Guo et al., 2021) is an encoder-only pretrained model, which extends the capabilities of CodeBERT (Feng et al., 2020) by incorporating a semantic-aware pre-training objective function. CodeT5 (Wang et al., 2021) employs T5 (Raffel et al., 2020) encoder-decoder architecture. In our implementations, we use CodeT5-base with 220M parameters. Here, we also investigate the behavior of larger models, including CodeT5+ (Wang et al., 2023) with 770M parameters and Code Llama with 13B parameters. CodeT5+ (Wang et al., 2023) is an extension of CodeT5 (Wang et al., 2021), and Code Llama (Rozière et al., 2023) is a decoder-only build on top of Llama 2 (Touvron et al., 2023) for code-specialized tasks. We provide more details in Appendix A.

**Downstream Tasks.** We study the behavior of the models on two different downstream tasks, including text-to-code generation and code refinement. These tasks are part of the most challenging tasks in the CodeXGLUE benchmark (Lu et al., 2021). **Text-to-code** is the task of generating a

program given a natural language description. In CodeXGLUE benchmark (Lu et al., 2021), CON-CODE dataset (Iyer et al., 2018) is proposed for this task. **Code refinement** is the task of resolving the bugs in a given program by automatically generating a corrected program. We use the medium dataset of Tufano et al. (2019).

**Evaluation Metrics.** Exact match (Wang et al., 2021), CodeBLEU (Ren et al., 2020), and BLEU score (Papineni et al., 2002) have been commonly used to evaluate the model performance in the downstream tasks. The exact match metric evaluates if the generated code matches the target code at the token-level. BLEU score measures the n-gram overlap between the output and the target code. CodeBLEU considers syntactic and data-flow matches of the codes in addition to the n-gram overlap. In this work, we focus on the exact match metric to quantify the model behaviors. This is due to the nature of OOD scenarios, where it is desirable to see if the model can generate specific unseen or rare programs correctly. It is important to note that Wang et al. (2021) have demonstrated that for the code refinement task, achieving a high BLEU score can be accomplished with a simple duplication of the input codes, comparable to state-of-the-art models. Furthermore, it has been shown that CodeBLEU and BLEU scores are not necessarily correlated with the correctness of the programs (Evtikhiev et al., 2022; Hendrycks et al., 2021). We report BLEU score results in Appendix F.

## 4.2 DATA CONSTRUCTION AND FINE-TUNING

In the data construction process, for each scenario, we choose $\tilde{\mathcal{P}}$ in a way that counts for $\approx 3\%$ of the entire fine-tuning data. In OOD scenarios, we mask out all of the data items with properties $\tilde{\mathcal{P}}$. For the few-data regime cases, we mask-out half (50%) of data with properties $\tilde{\mathcal{P}}$ ($\approx 1.5\%$ of the entire fine-tuning data). In all the scenarios, we infer the fine-tuned models on test data with $\tilde{\mathcal{P}}$ properties. Note that, in the text-to-code task, we mask out the data based on the target data (code data rather than text data) properties. For the code refinement tasks, we masked the data based on the input.

**Length-based Scenarios.** To generate data for length-based scenarios, we characterize the dataset of programs based on the token size. For each scenario, $\tilde{\mathcal{P}}$ specifies a continuous range of program token sizes. We consider five ranges in our experiments: $\tilde{\mathcal{P}}_1 = \{[0\%, 3\%]\}$, $\tilde{\mathcal{P}}_2 = \{[24\%, 27\%]\}$, $\tilde{\mathcal{P}}_3 = \{[48\%, 51\%]\}$, $\tilde{\mathcal{P}}_4 = \{[72\%, 75\%]\}$, and $\tilde{\mathcal{P}}_5 = \{[97\%, 100\%]\}$. Note that $\tilde{\mathcal{P}}_1 = \{[0\%, 3\%]\}$ represents the top 3% smallest programs, in terms of token size. We consider $\tilde{\mathcal{P}}_1$ and $\tilde{\mathcal{P}}_5$ as length-based extrapolation scenarios and $\tilde{\mathcal{P}}_2$, $\tilde{\mathcal{P}}_3$, and $\tilde{\mathcal{P}}_4$ as length-based interpolation scenarios.

**Syntax-based Scenarios.** In syntax-based scenarios, we characterize program datasets based on the histogram of language elements (e.g., syntax keyword *else*). For each task, we select five different language elements that cover $\approx 3\%$ of the fine-tuning data. For example, in text-to-code task we consider $\tilde{\mathcal{P}}_1 = \{else\}$. We provide details of the selected language elements in Appendix D.

**Semantic-based Scenarios.** In this work, we employ CodeT5+ (770M parameters) (Wang et al., 2023) encoder to characterize the semantics distribution of programs. We feed the tokenized programs to the CodeT5+ encoder and obtain the corresponding feature vectors $\mathbf{V}$ of size $1024 \times t$, where $t$ is the size of the input program. We obtain the continuous representation of the programs by averaging the tokens' embedding following (Koto et al., 2021). We then cluster the programs in continuous space using the K-means algorithm. We set the number of clusters $K = 35$ using the elbow method (Bholowalia & Kumar, 2014). To accelerate the clustering procedure, we perform dimensionality reduction PCA with a target dimension of 50. We determine the dimension in a way that all the components explain at least $80\%$ of the data variance. To have different semantic-based scenarios without losing generality, we randomly select five different clusters. Each cluster can represent a set of $\tilde{\mathcal{P}}_i$ properties. We provide examples of clusters' semantic in Appendix E.

**Model Fine-tuning Details.** We fine-tune four pretrained models for two different tasks in various scenarios. We stick to their defaults for fair comparisons. For fine-tuning the models with LoRA method, we follow the hyperparameters reported by Hu et al. (2022), where the pretrained weights are frozen, and we only optimize the injected rank decomposition matrices. We provide more details

Table 1: Overall results of the model performance for different scenarios in **text-to-code** task. The results provide the relative exact match to the 100% baseline (without data mask-out) for different OOD and few-data regime scenarios. Length Inter and Length Extra refer to length-based interpolation and extrapolation scenarios, respectively. FT denotes full fine-tuning, and LoRA refers to the LoRA fine-tuning method. OOD and Few refer to OOD and few-data regime scenarios, respectively.

| Models | | Length Inter | | Length Extra | | Syntax | | Semantic | |
|---|---|---|---|---|---|---|---|---|---|
| | | FT | LoRA | FT | LoRA | FT | LoRA | FT | LoRA |
| CodeT5 | OOD | 53.92% | 66.91% | 0.00% | 24.99% | 16.46% | 34.81% | 31.90% | 51.42% |
| | Few | 86.56% | 103.79% | 28.56% | 55.0% | 93.90% | 100.0% | 37.56% | 72.43% |
| CodeT5+ | OOD | 49.65% | 70.94% | 5.0% | 26.09% | 47.95% | 68.97% | 39.69% | 55.71% |
| | Few | 76.40% | 96.36% | 77.38% | 101.72% | 67.21% | 78.54% | 66.04% | 83.68% |
| Code Llama | OOD | - | 71.75% | - | 23.57% | - | 64.81% | - | 56.72% |
| | Few | - | 94.08% | - | 63.21% | - | 86.08% | - | 84.74% |

Table 2: Overall results of the model performance for different scenarios in **code refinement** task. The results provide the relative exact match to the 100% baseline (without data mask-out) for different OOD and few-data regime scenarios. Length Inter and Length Extra refer to length-based interpolation and extrapolation scenarios, respectively. FT denotes full fine-tuning, and LoRA refers to the LoRA fine-tuning method. OOD and Few refer to OOD and few-data regime scenarios, respectively. GCBERT denotes to the GraphCodeBERT model Guo et al. (2021).

| Models | | Length Inter | | Length Extra | | Syntax | | Semantic | |
|---|---|---|---|---|---|---|---|---|---|
| | | FT | LoRA | FT | LoRA | FT | LoRA | FT | LoRA |
| GCBERT | OOD | 82.91% | 87.89% | 37.82% | 74.35% | 1.30% | 2.35% | 60.38% | 69.05% |
| | Few | 86.52% | 94.45% | 90.15% | 90.46% | 75.42% | 77.92% | 76.45% | 84.43% |
| CodeT5 | OOD | 84.10% | 86.70% | 48.95% | 61.53% | 10.23% | 28.78% | 77.41% | 79.36% |
| | Few | 85.48% | 89.97% | 57.30% | 80.29% | 83.08% | 85.82% | 83.63% | 88.73% |
| CodeT5+ | OOD | 80.70% | 83.39% | 73.44% | 82.39% | 21.41% | 37.14% | 73.65% | 78.67% |
| | Few | 93.28% | 94.65% | 79.56% | 90.77% | 72.83% | 81.01% | 85.30% | 93.29% |
| Code Llama | OOD | - | 81.70% | - | 57.69% | - | 43.70% | - | 70.14% |
| | Few | - | 87.68% | - | 85.71% | - | 87.66% | - | 89.23% |

of the hyperparameters in Appendix B. All our experiments are conducted using a machine with four NVIDIA 40GB Ampere A100 GPUs.

## 4.3 HOW DO FINE-TUNED MODELS GENERALIZE IN OOD SCENARIOS?

Table 1 and Table 2 shows the overall results of different models in length-, syntax-, and semantic-based scenarios, respectively. These tables show the model performance in the OOD scenarios where the models do not have access to the fine-tuning data with $\tilde{\mathcal{P}}$ properties. Furthermore, Table 1 and Table 2 show how well the models perform when they have access to 50% of the masked data. Note that in Table 1 and Table 2, all of the results are the average of different scenarios and show the relative exact match to the 100% baseline (models with access to the full data distribution). In Table 1 and Table 2, we provide the results of fine-tuning the models using full fine-tuning and LoRA fine-tuning methods. Note that for Code Llama 13B, due to the substantial resource require-

Table 3: Exact match results of the fine-tuned models using the full fine-tuning dataset for text-to-code and code refinement tasks. FT denotes full fine-tuning, and LoRA refers to the LoRA fine-tuning method. GCBERT refers to Guo et al. (2021).

| Models | Text-to-Code | | Refinement | |
|---|---|---|---|---|
| | FT | LoRA | FT | LoRA |
| GCBERT | - | - | 10.74 | 11.38 |
| CodeT5 | 22.15 | 21.65 | 14.43 | 14.53 |
| CodeT5+ | 24.95 | 24.70 | 15.18 | 15.29 |
| Code Llama | - | 27.65 | - | 19.19 |

ments involved in full fine-tuning, we only report the LoRA fine-tuning results. Additionally, in line with GraphCodeBERT (Guo et al., 2021), we only investigate this model on the code refinement task. In these tables, for the length-based scenarios, we have five different scenarios, three for the interpolation cases and two for the extrapolation cases, so we report the average results for each case. In syntax-based and semantic-based scenarios, we report the average results of five different scenarios.

We conclude according to Table 1 and Table 2 that: 1. Interpolation cases in the length-based OOD scenarios, as we expected, are the easiest OOD scenarios for the models in different tasks. 2. Syntax-based and length-based extrapolation OOD scenarios are the most challenging scenarios for the models. 3. Using LoRA fine-tuning we can achieve significantly better generalization accuracy compared to full fine-tuning. 4. Few-data regime scenarios show that adding a few relevant data to the fine-tuning distribution can gain huge performance improvement. In the following, we describe our key findings in more detail.

**Model performance decreases in various OOD scenarios.** Table 1 and Table 2 show that all of the models have difficulty in dealing with different OOD scenarios. These include models with different architecture and parameter sizes. For example, in Table 1, we observe that for the Code Llama model with 13B parameters, the performance significantly dropped in the length-based extrapolation scenario. It achieves only 23.57% of the baseline performance (model with full data access).

Table 1 and Table 2 indicate that length-based interpolation scenarios are the least challenging OOD scenarios for various models in both text-to-code and code refinement tasks. While length-based interpolation is the easiest OOD scenario, it is worth noting that CodeT5+ with full fine-tuning only attains 49.67% of the baseline performance (See Table 1). Additionally, Table 1 and Table 2 reveal that the models exhibit the most significant performance reduction in the length-based extrapolation and syntax-based OOD scenarios. This performance drop occurred despite the models being exposed to similar examples during pre-training. For instance, the models had previously encountered code examples that included the *else* element.

A comparison between the outcomes of the semantic scenarios presented in Table 1 and Table 2 highlights that the text-to-code task is more challenging than the code refinement task. This is mainly due to the multi-modality nature of the task, wherein the models need to learn to map natural languages to the unseen or rare compositions of programs.

> **Takeaway:** Performance of fine-tuned models, regardless of architectures and parameter sizes, can significantly deteriorate in OOD scenarios, even when the models have seen similar code samples during pre-training.

**LoRA fine-tuning exhibits better OOD generalization compared to full fine-tuning.** In Table 1 and Table 2, we provide the results of fine-tuning the models using two different fine-tuning approaches: full parameter fine-tuning and LoRA fine-tuning. The results presented in these tables indicate that LoRA fine-tuning consistently exhibits superior OOD generalization across various scenarios. For example, Table 1 shows that in the length-based extrapolation scenario, fine-tuning CodeT5 with LoRA fine-tuning resulted in a 24.99% relative exact match, whereas the model's relative performance using full fine-tuning was 0.0%. Furthermore, as demonstrated in Table 2, in the syntax-based OOD scenario, the utilization of LoRA fine-tuning for fine-tuning CodeT5 and CodeT5+ results in significantly superior performance compared to employing full fine-tuning for these models. This observation shows that LoRA fine-tuning, which involves freezing the pretrained weights, effectively leverages the previously acquired knowledge, resulting in improved OOD generalization compared to full fine-tuning.

Table 3 provides in-distribution performance results of the models fine-tuned using both full fine-tuning and LoRA fine-tuning methods. This table displays the exact match accuracy of the models on the complete test set under the condition that the models have access to the entire fine-tuning distribution. Table 3 demonstrates that employing LoRA fine-tuning enables us to achieve performance that is comparable to full fine-tuning. It is important to highlight that in all of the experiments involving LoRA fine-tuning, the pretrained weights are frozen, and we only need to optimize newly injected weights. These LoRA parameters account for less than 1% of the pretrained weights. Note that we provide BLEU score results in Appendix C.

> **Takeaway:** While full fine-tuning and LoRA fine-tuning methods show comparable results over in-distribution data, LoRA fine-tuning significantly outperforms full fine-tuning in OOD scenarios. This suggests that with freezing pretrained weights, LoRA fine-tuned models can effectively utilize their pretraining knowledge in dealing with various OOD scenarios.

**Models can gain significant improvement by adding a few relevant data to the training set.** Table 1 and Table 2 provide the results for few-data regime scenarios. In these scenarios, we only mask out 50% of the data with $\tilde{\mathcal{P}}$ properties ($\approx 1.5\%$ of the fine-tuning data). The Table 1 and Table 2 demonstrate in each scenario that by adding data in size $\approx 1.5\%$ of the fine-tuning data in each specific scenario, the model can gain significant accuracy performance. For example, Table 1 shows that in syntax-based scenarios, applying LoRA fine-tuning to CodeT5 can lead to gain 100% of relative performance by adding a small amount of relevant data.

> **Takeaway:** By incorporating a small amount of relevant data (representing $\approx 1.5\%$ of the fine-tuning data) into the fine-tuning set, models can achieve substantial performance enhancements.

### 4.4 CAN FINE-TUNED LLMS GENERATE UNSEEN LANGUAGE ELEMENTS?

In the syntax-based OOD scenarios, we can assess the fine-tuned LLMs' ability to leverage their prior knowledge in generating unseen language elements. For instance, can the fine-tuned models generate the *else* element if they have not been exposed to any code data containing *else* during fine-tuning? In Figure 3, we present the relative frequencies of generating unseen elements by models fine-tuned using both full fine-tuning and LoRA fine-tuning methods. The results in Figure 3 show the frequencies of generating unseen elements relative to the frequencies in ground truth programs. We report the average results of five different unseen elements during fine-tuning. The list of these elements was reported Appendix D. In Figure 3, the solid bars represent the results for models fine-tuned using full fine-tuning, while the hatched bars depict the results for models fine-tuned using the LoRA method.

Figure 3 shows that the fine-tuned LLMs are able to generate unseen language elements in different tasks. More interestingly, the models fine-tuned using the LoRA fine-tuning exhibit the ability to generate a higher percentage of unseen elements when compared to models fine-tuned with the full fine-tuning approach. This indicates that the models fine-tuned with the LoRA method possess a superior capability to leverage their previously acquired knowledge and are less prone to forgetting. Furthermore, we observe that generating unseen language elements is more challenging in the text-to-code task (Figure 3a) compared to the code refinement task (Figure 3b). The main reason is that in the text-to-code task, the models

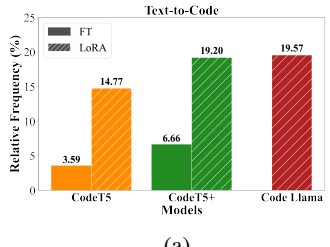

(a)

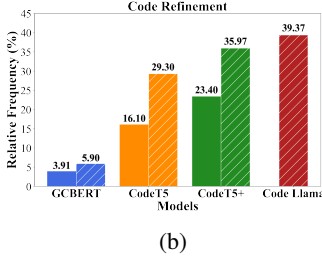

(b)

Figure 3: The ratios of frequency of generated unseen language elements over the frequency in ground truth data. Solid and hatched bars show the results of the model fine-tuned with the full fine-tuning and LoRA fine-tuning, respectively.

need to learn the mapping from natural language to the programs. As shown in Figure 3b, Graph-CodeBERT (Guo et al., 2021) generates a lower number of unseen elements. This phenomenon is primarily attributed to the fact that in encoder-only models, we initialize the decoder from scratch. Consequently, these models can only rely on the prior knowledge acquired by the encoder.

> **Takeaway:** Our findings indicate that models fine-tuned with LoRA generate a greater number of unseen elements compared to those fine-tuned using the full fine-tuning. This suggests that LoRA fine-tuned models exhibit an enhanced ability to leverage their pretraining knowledge when compared to models fine-tuned with full fine-tuning.

## 5 CONCLUSION

In this work, we propose a systematic approach to investigate the behaviors of fine-tuned LLMs in OOD scenarios for the source code domain. Given the data, we simulate OOD scenarios based on the program's length, syntax, and semantics. Using the simulated scenarios, we shed light on the models' fragility in the OOD scenarios, potential performance drop, and the necessity to improve dataset construction. We also reveal the model's impotence in handling considered OOD dimensions

and to what extent we can improve the generalization of the models by exposing the relevant data. Furthermore, our results reveal that, although models fine-tuned with full fine-tuning and LoRA exhibit similar in-distribution accuracy, LoRA fine-tuning demonstrates higher OOD generalization accuracy. We will release the implementation of our work to support future systematic analysis of source code model behavior in OOD scenarios.

## REPRODUCIBILITY STATEMENT

We provide the details of data construction and fine-tuning procedure in subsection 4.2. This covers the details of data construction for each scenario and explains how we utilized a pretrained encoder to define the semantic scenarios. Additionally, we also provide the detail of the pretrained models we examined in Appendix A. In all of the scenarios, to fine-tune the GraphCodeBERT (Guo et al., 2021), CodeT5 (Wang et al., 2021), and CodeT5+ (Wang et al., 2023) using the full fine-tuning method, we follow the provided hyperparameters by the authors of each work. The hyperparameters of LoRA fine-tuning, including the hyperparameters for fine-tuning Code Llama (Rozière et al., 2023), are provided in Appendix B. Furthermore, we provide the details of the selected language elements of our syntax-based scenarios in Appendix D. At the time of publication, we will publish the implementation of our work.

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

# A    PRETRAINED MODELS

Here, we provide more detail about the pretrained models we used in our experiments.

## A.1    BERT-BASED MODELS

CodeBERT (Feng et al., 2020) is an encoder-only transformer-based model that is pretrained using CodeSerchNet dataset (Husain et al., 2019). This dataset consists of 2.1M pairs of individual functions and code documentations with 6.4M code-only data items across multiple programming languages. This model uses a 12-layer RoBERTa-based (Liu et al., 2019) architecture with 125M parameters. It is trained using masked language modeling (MLM) and replaced token detection objective.

Guo et al. (2021) proposed GraphCodeBERT by extending CodeBERT (Feng et al., 2020) using a semantic-aware pre-training objective function. They incorporate data-flow information in the pre-training stage to encode the semantic information of the program.

## A.2    CODET5

CodeT5 (Wang et al., 2021) employ T5 (Raffel et al., 2020) encoder-decoder architecture. The authors use CodeSearchNet (Husain et al., 2019) with 1.2M pairs of functions' code with corresponding documentation, and 0.8M code-only data items. In our experiments, we use CodeT5-base with 220M. This model uses MLM objective and identifier-aware objective functions in the pre-training procedure.

CodeT5+ (Wang et al., 2023) is a family of encoder-decoder LLMs (Wang et al., 2021) that is developed with the flexibility to cover a wide range of downstream tasks. CodeT5+ achieved this flexibility by employing a mixture of pretraining objectives including span denoising, contrastive learning, text-code matching, and causal LM pretraining tasks(Wang et al., 2023). In our experiments we employ CodeT5+ with 770M parameters.

## A.3    CODE LLAMA

Code Llama (Rozière et al., 2023) is a family of LLM for code developed based on Llama 2 models (Touvron et al., 2023). The models are designed using decoder-only architectures with 7B, 13B, and 34B parameters. Code Llama encompasses different versions tailored for a wide array of tasks and applications, including the foundational model, specialized models for Python code, and instruction-tuned models. Code Llama outperforms open models on HumanEval (Chen et al., 2021) and MBPP benchmarks (Austin et al., 2021) up to 53% and 55%, respectively. In our experiments, we use the foundation model version of Code Llama with 13B parameters.

# B    HYPERPARAMETERS FOR LoRA FINE-TUNING

In Table 4, we present the LoRA hyperparameters that were applied in the fine-tuning of various models. We fine-tune these models utilizing AdamW with a linear learning rate decay schedule. During the validation and testing phases, we employed beam search with a beam size of 10, following Wang et al. (2021; 2023); Guo et al. (2021).

For fine-tuning GCBERT, CodeT5, and CodeT5+ in the text-to-code task, we set the maximum input and output sequence length to 320 and 150 tokens, respectively. In the case of fine-tuning Code Llama, we set the maximum sequence length to 470 tokens. In the code refinement task, to fine-tune GCBERT, CodeT5, and CodeT5+, we set the maximum input and output sequence length to 240 and 240 tokens. We fine-tune Code Llama for code refinement tasks by setting the maximum sequence length to 480.

Table 4: The LoRA hyperparameters we used to fine-tune the models for text-to-code and code refinement tasks.

| Models | Batch Size | #Epoch | Learning Rate | Rank $(r_q, r_v)$ | LoRA $\alpha$ |
|---|---|---|---|---|---|
| GCBERT | 32 | 20 | $5e^{-4}$ | 16, 16 | 32 |
| CodeT5 | 32 | 20 | $5e^{-4}$ | 16, 16 | 32 |
| CodeT5+ | 16 | 15 | $5e^{-4}$ | 16, 16 | 32 |
| Code Llama | 4 | 5 | $5e^{-4}$ | 16, 16 | 32 |

Table 5: Exact match (EM) and BLEU (B) results of the fine-tuned models using the full fine-tuning dataset for text-to-code and code refinement tasks. FT denotes full fine-tuning, and LoRA refers to the LoRA fine-tuning method. GCBERT refers to Guo et al. (2021).

| Models | Text-to-Code | | | | Refinement | | | |
|---|---|---|---|---|---|---|---|---|
| | FT | | LoRA | | FT | | LoRA | |
| | EM | B | EM | B | EM | B | EM | B |
| GCBERT | - | - | - | - | 10.74 | 90.93 | 11.38 | 86.45 |
| CodeT5 | 22.15 | 39.60 | 21.65 | 38.90 | 14.43 | 89.33 | 14.53 | 89.40 |
| CodeT5+ | 24.95 | 44.06 | 24.70 | 43.78 | 15.18 | 88.19 | 15.29 | 89.65 |
| Code Llama | - | - | 27.65 | 45.19 | - | - | 19.19 | 90.34 |

## C  COMPARISON OF FULL FINE-TUNING AND LoRA FINE-TUNING METHOD

In Table 5, you can find the in-distribution performance results of fine-tuned models using the full and LoRA fine-tuning methods. This table corresponds to a version of Table 3, which additionally includes BLEU score results.

## D  LIST OF LANGUAGE ELEMENTS

In syntax-based scenarios, we consider one element in each scenario and mask-out the source code with that particular element. Here, we provide the details of five language elements used in our experiments. Note that we pick the element that covers $\approx 3\%$ of the fine-tuning data. We conduct our syntax-based experiments based on the following language elements of each task,

1. **Text-to-Code**: {*else*, *floating_point_type*, *unary_expression*, *array_access*, *true*}
2. **Code Refinement**: {*while_statement*, *long*, *array_creation_expression*, *break*, $\geqslant$}

## E  DO THE CLUSTERS REPRESENT PROGRAMS WITH SPECIFIC SEMANTICS?

Table 6 provides semantics of five random clusters (out of 35) in text-to-code tasks. We randomly check 20 source codes in each cluster to check their semantics.

Table 6: Semantics of five clusters in text-to-code task.

| Cluster-ID | Semantic |
|---|---|
| 0 | Property setter functions |
| 1 | Property string getter functions |
| 6 | Initialize object |
| 11 | Using getter function |
| 17 | String concatenation |

Table 7: Overall results of the model performance for different scenarios in **text-to-code** task. The results provide the BLEU score for different scenarios. Length Inter and Length Extra refer to length-based interpolation and extrapolation scenarios, respectively. FT denotes full fine-tuning, and LoRA refers to the LoRA fine-tuning method. OOD and Few refer to OOD and few-data regime scenarios, respectively. Full refers to 100% baseline (when a model has access to 100% of the fine-tuning set).

| Models | | Length Inter | | Length Extra | | Syntax | | Semantic | |
|---|---|---|---|---|---|---|---|---|---|
| | | FT | LoRA | FT | LoRA | FT | LoRA | FT | LoRA |
| | OOD | 40.19 | 42.03 | 15.09 | 15.23 | 24.08 | 24.18 | 44.58 | 46.21 |
| CodeT5 | Few | 48.91 | 46.47 | 20.18 | 18.46 | 25.20 | 24.95 | 45.43 | 47.97 |
| | Full | 47.79 | 48.34 | 24.08 | 23.34 | 27.01 | 25.83 | 48.48 | 49.65 |
| | OOD | 40.58 | 44.07 | 15.98 | 17.48 | 24.39 | 26.41 | 40.52 | 43.11 |
| CodeT5+ | Few | 50.07 | 50.10 | 19.33 | 21.67 | 27.25 | 27.25 | 48.93 | 50.77 |
| | Full | 51.80 | 51.23 | 23.29 | 22.63 | 28.98 | 28.04 | 50.89 | 51.03 |
| | OOD | - | 54.34 | - | 21.24 | - | 25.37 | - | 47.74 |
| Code Llama | Few | - | 60.35 | - | 36.73 | - | 28.06 | - | 50.76 |
| | Full | - | 62.11 | - | 37.44 | - | 29.50 | - | 51.38 |

Table 8: Overall results of the model performance for different scenarios in **code refinement** task. The results provide the BLEU score for different scenarios. Length Inter and Length Extra refer to length-based interpolation and extrapolation scenarios, respectively. FT denotes full fine-tuning, and LoRA refers to the LoRA fine-tuning method. OOD and Few refer to OOD and few-data regime scenarios, respectively. Full refers to 100% baseline (when a model has access to 100% of the fine-tuning set). GCBERT denotes to the GraphCodeBERT model Guo et al. (2021).

| Models | | Length Inter | | Length Extra | | Syntax | | Semantic | |
|---|---|---|---|---|---|---|---|---|---|
| | | FT | LoRA | FT | LoRA | FT | LoRA | FT | LoRA |
| | OOD | 88.22 | 88.37 | 83.01 | 81.45 | 79.44 | 81.74 | 88.36 | 85.76 |
| GCBERT | Few | 88.59 | 88.32 | 85.14 | 82.75 | 90.36 | 87.67 | 88.95 | 86.28 |
| | Full | 88.32 | 88.56 | 84.61 | 82.99 | 90.10 | 87.93 | 89.73 | 86.45 |
| | OOD | 87.37 | 88.65 | 80.35 | 84.11 | 83.05 | 87.08 | 84.68 | 87.75 |
| CodeT5 | Few | 86.67 | 88.06 | 81.62 | 84.22 | 89.19 | 90.19 | 86.54 | 88.24 |
| | Full | 87.39 | 88.74 | 83.22 | 84.22 | 89.88 | 88.78 | 87.69 | 88.96 |
| | OOD | 83.08 | 86.29 | 81.26 | 82.15 | 84.60 | 85.48 | 84.73 | 85.97 |
| CodeT5+ | Few | 84.81 | 87.30 | 83.03 | 82.26 | 88.83 | 88.96 | 85.91 | 86.72 |
| | Full | 86.05 | 87.75 | 83.17 | 83.16 | 89.45 | 89.01 | 87.46 | 86.62 |
| | OOD | - | 86.40 | - | 78.30 | - | 83.29 | - | 81.32 |
| Code Llama | Few | - | 88.79 | - | 84.07 | - | 90.92 | - | 89.12 |
| | Full | - | 89.03 | - | 84.26 | - | 91.96 | - | 89.80 |

# F MORE EXPERIMENTAL RESULTS

## F.1 BLEU SCORE RESULTS

In Table 7 Table 8, we provide BLEU score results of different scenarios for the text-to-code and code refinement tasks, respectively. As we mention in subsection 4.1, BLEU scores are not necessarily correlated with the correctness of the programs (Hendrycks et al., 2021) and human judgment (Evtikhiev et al., 2022). For example, a text-to-code model with a high BLEU score could mislead users. Furthermore, Wang et al. (2021) show that in the code refinement task, the BLEU score of a naive copy of the input code can be as good as the state-of-the-art methods. Table 7 shows the performance (BLEU score) dropped for different models in all of the OOD scenarios compared to the 100% baseline. For example, in the length-based extrapolation scenario for the CodeLlama model, the BLEU score dropped over 16 points when compared to the 100% baseline performance. Fur-

**Input text:** Returns true if view's layout direction is right-to-left.

(a) Target Code

```
1 boolean function (View arg0) {
2   if ( Build.VERSION.SDK_INT >=
        VERSION_CODES.JELLY_BEAN_MR1 )
          {
3   return arg0.getLayoutDirection()
        == View.LAYOUT_DIRECTION_RTL;
4   }
5   else {
6   return false;
7   }
8 }
```

(b) Generated Code

```
1
2 boolean function (View arg0) {
3   return arg0.getLayoutDirection()
          == View.LAYOUT_DIRECTION_RTL;
4 }
```

Figure 4: An example of generated code by Code Llama in the syntax-based OOD scenario for the text-to-code task. Here *else* is the unseen language element.

thermore, as shown in Table 7, it is evident that across all OOD scenarios, fine-tuning the models using the LoRA approach consistently results in higher BLEU scores. As depicted in Table 8, it is apparent that there are fewer performance drops in comparison to the text-to-code results outlined in Table 7. This distinction can be primarily attributed to the code refinement task's inherent characteristics, wherein naively copying the input tokens to the outputs can yield state-of-the-art BLEU scores.

## F.2 QUALITATIVE EXAMPLES

In Figure 4, Figure 5, and Figure 6, we present qualitative results showcasing instances where the Code Llama model was not able to generate the targeted codes in the OOD scenarios. These examples highlight the challenge that even large fine-tuned LLMs face when handling OOD data. Figure 4 shows an example of the syntax-based OOD scenarios in which the model was unable to generate and use the *else* element. In Figure 5 demonstrates another example from the text-to-code task. Here, we provide an example of the length-based extrapolation OOD scenarios. In these scenarios, our goal is to investigate whether the model is able to extrapolate from shorter programs to longer ones. Figure 5 shows that Code Llama was unable to generate the target program correctly. Note that Figure 5 shows an example of $\tilde{\mathcal{P}}_5 = \{[97\%, 100\%]\}$ OOD scenario, where only 3% of the entire fine-tuning data is masked out. Figure 6 shows an example of the code refinement task. In Figure 6, we provide an example of the syntax-based scenario, in which Code Llama encountered difficulty in generating the *while_statement*. In this syntax-based scenario *while_statement* is the unseen language element.

**Input text:** Does this nodetest pass using the specified nodetester instance?

(a) Target Code

```
1 void function (NodeTester arg0,
      short [ ] arg1) {
2   NodeIterator loc0 =
      documentTraversal.
      createNodeIterator(rootNode ,
      NodeFilter.SHOW_ALL, new
      NodeTypeNodeFilter(arg1), true
      );
3   for(Node loc1 = loc0.loc1 ( );
      loc1 != null; loc1 = loc0.loc1
      ()){
4     arg0.testNode(loc1, this);
5   }
6   arg0.noMoreNodes(this);
7   }
```

(b) Generated Code

```
1 String function (Node arg0) {
2   StringBuilder loc0 = new
      StringBuilder();
3   handleWsRec(arg0, loc0);
4   return loc0.toString();
5 }
```

Figure 5: An example of generated code by Code Llama in the length-based OOD scenario for the text-to-code task. Here we provide an example of length-based extrapolation scenarios ($\tilde{\mathcal{P}}_5 = \{[97\%, 100\%]\}$).

(a) Target Code

```
1  public java.lang.String toString(){
2   java.lang.String VAR_1 = STRING_1;
3   int temp = start;
4   while ( temp != end){
5     VAR_1 += (array[temp]) + STRING_2
        ;
6     if( temp == (( array.length ) - 1
        )){
7       temp = 0;
8     }else{
9       temp ++;}
10  }
11  VAR_1 += array[temp];
12  VAR_1 += STRING_3;
13  return VAR_1;
14  }
```

(b) Generated Code

```
1  public java.lang.String toString(){
2   java.lang.String VAR_1 = STRING_1;
3   if ((METHOD_1()) != null ){
4     VAR_1 += METHOD_2();
5     if (( METHOD_1 ()) != null){
6       VAR_1 += STRING_2;
7     }
8   }
9   VAR_1 += STRING_3;
10  return VAR_1;
11  }
```

Figure 6: An example of generated code by Code Llama in the syntax-based OOD scenario for the code refinement task. Here *while_statement* is the unseen language element.).

