# OpenReview forum: "SimSCOOD: Systematic Analysis of Out-of-Distribution Generalization in Fine-tuned Source Code Models"
_ICLR.cc/2024/Conference — ICLR 2024 Conference Withdrawn Submission_

### Official Review · Reviewer_fJQ1 · 2023-10-19

**Soundness:** 3 good
**Presentation:** 3 good
**Contribution:** 2 fair
**Rating:** 5
**Confidence:** 4

**Summary:**

The paper conducts a systematic investigation of the behaviors of code models under different fine-tuning techniques. They create diverse OOD scenarios by masking out some portion of distributions. Then the model is either fine-tuned fully or partially (via LoRa). They reveal some important insights into the OOD generalization ability of LoRa and FT.

**Strengths:**

1. The OOD scenarios are comprehensive. The authors propose three kinds of OOD simulations, including length-based, syntax-based, and semantics-based methods. These scenarios are sufficient to cover real-life OOD data.

2. The behavior analyses of the behavior models are thorough and systematic. The authors compare full-finetuning and LoRa under different test benchmarks. Sufficient insights and conclusions are given based on the analysis of the results.

**Weaknesses:**

1. The main weakness to me is that all the conclusions and insights are somehow predictable. It is somehow natural that LoRa generalizes better than FT as the fine-tuned weights are fewer and OOD data are more scarce than the pre-training data. Other insights are also similar: we can get them by logically reasoning about the approach. The only useful takeaway to me is the specific performance gain of introducing extra data.

2. As discussed above, the specific performance gain brought by extra data is very interesting to me. Could you conduct an ablation study to show the performance gain versus different amounts of extra data? This would give more valuable clues to the community as we would know using how much data for fine-tuning would be the best tradeoff.

3. Can you conduct an ablation study of increasing the amount of fine-tuning data and the fine-tuning time? This might decrease the advantage of LoRa against FT.

Finally, as a minor comment, I noticed that the authors consistently misuse \citet and \citep throughout the whole paper. I suggest the authors proofread the paper a few more times to improve the readability.

**Questions:**

Please see the weaknesses.

---

> ### Author Response · Authors · 2023-11-15
>
> ### **The main weakness to me is that all the conclusions and insights are somehow predictable.**
>
> We addressed this concern in the second paragraph of **General response**. We hope our response covered the concerns. Please let us know if we have to provide more information about that.
>
> ### **Could you conduct an ablation study to show the performance gain versus different amounts of extra data? This would give more valuable clues to the community as we would know using how much data for fine-tuning would be the best tradeoff.**
>
> Thank you for the insightful suggestion. We are working on the suggested experiments and try our best to provide the results before the end of the discussion period.
>
> ### **Can you conduct an ablation study of increasing the amount of fine-tuning data and the fine-tuning time? This might decrease the advantage of LoRa against FT.**
>
> Thanks for the suggestion. We provide the fine-tuning time of above experiments for different extra data (E.g., presenting the fine-tuning time of adding an extra amount of masked data). Please let us know if the question mainly asked for comparing LoRA and FT's running time with different numbers of data (e.g., 1k, 10k, 100k).
>
> ### **I noticed that the authors consistently misuse \citet and \citep throughout the whole paper. I suggest the authors proofread the paper a few more times to improve the readability.**
>
> Thank you for bringing this to our attention. We had a pass over the paper and resolved the issue with \citet and \citep. The new version of the paper is uploaded.

---

> ### Author Response · Authors · 2023-11-22
>
> Here, we provide the results for the following question. We hope that the description and the results resolve your concerns.
>
> ### **Could you conduct an ablation study to show the performance gain versus different amounts of extra data?**
>
> In Table-1 and Table-2 we show the effect of revealing different percentages of the masked data on the model's performance. Specifically, we showcase CodeT5+ performance in different scenarios by revealing 25%, 50%, and 75% of the masked data (The data was masked for the OOD scenarios). Table-1 presents results for the text-to-code task, while Table-2 displays results for the code refinement task.
>
>
>
> Table-1 and Table-2 demonstrate that the model can gain a high performance even by revealing 25% (~ 0.75% of training data). For instance, in Table-1, within length extrapolation scenarios, the full fine-tuned model notably showed relative performance increases from 5.0% (OOD) to 64.63% (Few-25%). Furthermore, both tables indicate that revealing 50% and 75% of the masked data can enhance the model's performance across different scenarios. Nevertheless, the observed performance gains for Few-75% are less apparent compared to the Few-50% and Few-25% cases.
>
>
> **Table-1:** Overall model performance results for different scenarios with different amounts of data in **text-to-code** task. Few-XX% show the results of revealing 25%, 50%, and 75% of the masked data to the model. Note that OOD and Few-50% were presented in the paper.
> | | Length Inter | Length Extra | Syntax | Semantic|
> | ---------- | ---- | -- | ---- | ---- |
> | | *FT ............ LoRA*| *FT ............ LoRA* | *FT ............ LoRA* | *FT ............ LoRA*|
> | OOD | 49.65% ... 70.94% | 5.0%  ... 26.09% | 47.95% ... 68.97% |39.69% ... 55.71% |
> | Few-25%  | 69.34% ... 88.72% | 64.63% ... 86.55% | 63.16% ... 73.75%| 59.71% ... 78.47% |
> | Few-50% | 76.40% ... 96.36% | 77.38% ... 101.72% | 67.21% ... 78.54%  | 66.04% ... 83.68%|
> | Few-75% | 89.32% ... 98.82%  | 93.62% ... 99.36% | 79.50% ... 88.73%  | 76.65% ... 91.28% |
>
>
>
>
>
> **Table-2:** Overall model performance results for different scenarios with different amounts of data in **code refinement** task. Few-XX% show the results of revealing 25%, 50%, and 75% of the masked data to the model. Note that OOD and Few-50% were presented in the paper.
> | | Length Inter | Length Extra | Syntax | Semantic|
> | ---------- | ---- | -- | ---- | ---- |
> | | *FT ............ LoRA*| *FT ............ LoRA* | *FT ............ LoRA* | *FT ............ LoRA*|
> | OOD | 80.70% ... 83.39% | 73.44% ... 82.39% | 21.41% ... 37.14% | 73.65% ... 78.67% |
> | Few-25%  | 89.66% ... 91.53% | 76.82% ... 87.47% | 58.36% ... 75.44%| 81.48% ... 88.82% |
> | Few-50% | 93.28% ... 94.65% | 79.56% ... 90.77% | 72.83% ... 81.01% | 85.30% ... 92.29%|
> | Few-75% | 98.23% ... 99.51%  | 86.56% ... 92.21% | 84.24% ... 89.75%  | 89.32% ... 96.52% |

---

> ### Author Response · Authors · 2023-11-22
>
> We try our best to provide the requested ablation study. Please let us know if we need to provide any other experimental results. We appreciate a short reply to let us know your thoughts. Thanks a lot for your time.
>
> ###  **Can you conduct an ablation study of increasing the amount of fine-tuning data and the fine-tuning time?**
> In Table-1, we present CodeT5+ fine-tuning times for various data quantities, specifically focusing on the fine-tuning time for both full fine-tuning and LoRA fine-tuning over **10 epochs**. The results showcase the fine-tuning time using 1k, 10k, and 100k data. Overall, Table-1 indicates that full fine-tuning exhibits a slower fine-tuning time compared to LoRA fine-tuning across varying amounts of data.
>
> **Table-1:** Fine-tuning time (in **second**) of CodeT5+ model for text-to-code task using different number of data. **FT** refers to full fine-tuning method, and **LoRA** denotes LoRA fine-tuning model.
> |  | 1k | 10k | 100k |
> | ---------- | ---- | -- | ---- |
> | FT |  435s|  3761s| 35312s |
> | LoRA  | 253s | 2144s | 20326s |

---

### Official Review · Reviewer_9kMw · 2023-10-28

**Soundness:** 2 fair
**Presentation:** 3 good
**Contribution:** 1 poor
**Rating:** 5
**Confidence:** 4

**Summary:**

The paper analysis the out-of-distribution performance and forgetting phenomenon of source code models.
They mainly test several code models with full-finetuning and LoRA methods and give some conclusions based on the experiments results.

**Strengths:**

1. The writing and presentation are fluent.
2. The experiment results are significant when comparing the two methods.

**Weaknesses:**

1. The paper mainly tests several existing methods for the forgetting task of code datasets.
The novelty of this work is not clear from the conclusion and experiments now.

2. As an experimental paper, it lacks enough comparison of methods that prevent forgetting, e.g., Wise-FT[1].

3. For analyzing the forgetting phenomenon, there lacks the analysis on different hyperparameters' choice of finetuning
or the size of the dataset.

[1].Wortsman, Mitchell, et al. "Robust fine-tuning of zero-shot models. 2022 IEEE." CVF Conference on Computer Vision and Pattern Recognition (CVPR). 2021.

**Questions:**

1. It will be better to add more experiments mentioned above to formalize an extensive analysis.
2. The novelty of the work should be clearer.

---

> ### Author Response · Authors · 2023-11-15
>
> Thank you for the time taken to review our work. We hope we can address your concerns.
>
> ### **The paper mainly tests several existing methods for the forgetting task of code datasets. The novelty of this work is not clear from the conclusion and experiments now.**
>
> Note that the main focus of the paper is not forgetting issues. We are mainly studying the OOD issues of the pre-trained source code models, where forgetting issues can be a subset of OOD issues in our scenarios. For example, a model can perform poorly in the semantic scenario when it does not see similar examples during pre-training or fine-tuning.
>
> About the **novelty** of the work: to the best of our knowledge, this is the **first work** that studies the OOD behavior of large language models (e.g., Code Llama and CodeT5+) in generation tasks. To do that, we propose a systematic approach and, for the first time, define OOD scenarios for source code data along three code-related diverse dimensions: length, syntax, and semantic. Our results along each dimension raise awareness about the potential OOD issues of full fine-tuned and LoRA fine-tuned source code models.
>
> ### **As an experimental paper, it lacks enough comparison of methods that prevent forgetting, e.g.Wise-FT.**
>
> Thanks a lot for your suggestion. We agree that there are different methods to mitigate catastrophic forgetting. However, in this work, our main focus is to show the performance of two widely used fine-tuning approaches in the source code OOD scenarios. We show that regardless of architectures and parameter sizes, the models' performance dropped in various OOD scenarios (Even for Code Llama 6B, which was fine-tuned using the LoRA method). The paper's main message is to raise awareness about the OOD issues of the fine-tuned LLMs in the source code domain.
>
> ### **For analyzing the forgetting phenomenon, there lacks the analysis on different hyperparameters' choice of finetuning or the size of the dataset.**
>
> In our study, for a fair comparison with the baselines, we use the hyperparameters' of previous works (We specified these hyperparameters in the paper and our Reproducibility Statement), and we also used two well-known datasets with different sizes. As mentioned above, our main goal was to investigate the performance of two fine-tuning methods in OOD scenarios. However, we think analyzing different hyperparameters' choices would be an interesting future study.

---

### Official Review · Reviewer_cCst · 2023-10-31

**Soundness:** 2 fair
**Presentation:** 3 good
**Contribution:** 2 fair
**Rating:** 5
**Confidence:** 3

**Summary:**

This paper investigates the behavior of code Large Language Models (LLMs) fine-tuned in a source code domain when faced with out-of-distribution (OOD) code scenarios in the testing phase. The authors introduce a systematic approach to probe this issue. They begin by creating three types of OOD scenarios with modifications along various dimensions. Subsequently, they evaluate four leading code LLMs using both full fine-tuning and Low-Rank Adaptation (LORA) fine-tuning for two code-related tasks within these scenarios. The key findings highlight the challenges associated with OOD generalization in current code LLMs, the potential benefits of even a small amount of relevant labeled data, and the superiority of LORA fine-tuning in preserving the generalization capabilities of LLMs over full fine-tuning.

**Strengths:**

1) The problem studied is highly meaningful and significant. This paper pioneers the investigation of OOD generalization issues in code LLMs.

2) The experiments are extensive, as evidenced by the creation of diverse OOD code scenarios and the evaluation of various state-of-the-art code LLMs.

3) The paper is well-organized and easy to follow, with the key takeaways providing a concise summary of the empirical study's results.

**Weaknesses:**

1) The findings from the code data do not appear to be surprising and align with similar observations in prior studies involving OOD scenarios with image data. As the authors themselves acknowledge, previous works, such as Kumar et al. (2022), have already shown that full fine-tuning tends to exhibit weaker OOD generalization performance compared to fine-tuning with partial parameters. Additionally, the positive impact of a small amount of labeled data on generalization is in line with expectations, given the motivations within the realm of few-shot learning.

2) Do the three types of code OOD scenarios accurately represent realistic code OOD data? If future research builds upon this empirical study to enhance OOD performance in these three scenarios, can it be assured that the methods will also improve OOD performance in various other OOD scenarios?

3) The experimental details have been mentioned several times in various paragraphs. I would kindly recommend refining the presentation to make it more concise and informative.

**Questions:**

Kindly refer to Weaknesses for all questions.

---

> ### Author Response · Authors · 2023-11-15
>
> ### **The findings from the code data do not appear to be surprising and align with similar observations in prior studies involving OOD scenarios with image data.**
>
> We addressed this concern in the second paragraph of **General response**. We hope our response covered the concerns. Please let us know if we have to provide more information about that.
>
> ### **Do the three types of code OOD scenarios accurately represent realistic code OOD data?**
>
> We also address this question in the second part of **General response** (Paragraph 6).
> ### **The positive impact of a small amount of labeled data on generalization is in line with expectations, given the motivations within the realm of few-shot learning.**
>
> We agree that the output of using a few examples might be intuitive. However, before this study, we did not know how the models in each OOD scenario could benefit from the small amount of data. For example, in Table-1 we show that adding a small amount of relevant data to fine-tuning data for CodeT5 can lead to a gain of ~100% of relative performance. Therefore, our diverse scenarios and the few-data regime analysis provide insight toward shaping future datasets.
>
>
>
> ### **The experimental details have been mentioned several times in various paragraphs. I would kindly recommend refining the presentation to make it more concise and informative.**
>
> Thank you for bringing this to our attention. We will have another pass over the paper and try to remove any repetitive information before the end of the discussion period. It would be great help if you can point out the specific part of the paper that was repetitive.

---

### Official Review · Reviewer_HtnZ · 2023-11-04

**Soundness:** 3 good
**Presentation:** 3 good
**Contribution:** 2 fair
**Rating:** 5
**Confidence:** 3

**Summary:**

This paper investigates the out-of-distribution (OOD) generalization issue of fine-tuning pre-trained source code models. It contributes a systematic approach that simulates various OOD scenarios along different dimensions of source code data properties, including the length, syntax, and semantics. It then investigates the behaviors of models under different fine-tuning methodologies, including full fine-tuning and Low-Rank Adaptation (LoRA). The analysis is conducted on four state-of-the-art pre-trained models and applied to two code generation tasks, which exposes multiple failure modes attributed to OOD generalization issues and shows that LoRA fine-tuning consistently exhibits better OOD generalization performance than full fine-tuning.

**Strengths:**

* This paper contributes a systematic approach to simulate various OOD scenarios along different dimensions of source code data properties, including the properties of the length, syntax, and semantics.

* This paper studies the fine-tuned model behaviors in OOD scenarios on four pre-trained models, two tasks, and two fine-tuning methods and gives some takeaway conclusions, which may encourage future studies on this topic.

**Weaknesses:**

* The finding that fine-tuning may distort the pre-trained features, cause catastrophic forgetting, and harm the OOD generalization performance has already been studied in various NLP and CV tasks, as mentioned in RELATED WORK of this paper. This paper only changes to a new task of code generation, performs similar analysis, and gets similar results compared with existing works, which does not give much new insights for the community.

* There are some issues with the approach to simulate OOD scenarios in this paper. (1) The OOD scenarios made by masking out sub-regions of data distributions may not be realistic, especially in the cases of different lengths and certain language elements. (2) This paper also mentions the works of OOD analysis in programming languages in RELATED WORKS, and their relationships with this paper should be discussed in more detail. Why not use their pre-defined scenarios for evaluation? Would the scenarios in those works be more realistic?

* There are many methods proposed in CV and NLP to mitigate catastrophic forgetting in fine-tuning, but only the LoRA method is evaluated, which makes the experimental results not comprehensive enough.

----------------------------------------------
**Post Rebuttal:**

I thank the authors for providing the responses, which address some of the concerns and weaknesses in the review. The remaining concern is that the OOD generalization analysis in code generation, although claimed to be first studied in this paper, does not give some new insights compared with existing OOD generalization works. It would be more interesting and insightful if the analysis could find new results and conclusions specific to code generation.

**Questions:**

In Section 4.4, what is the performance of the original pre-trained model without FT or LoRA to generate unseen language elements?

---

> ### Author Response · Authors · 2023-11-15
>
> ### **The finding that fine-tuning may distort the pre-trained features, cause catastrophic forgetting has already been studied in various NLP and CV tasks**
>
> We addressed this concern in the second paragraph of **General response**. We hope our response covered the concerns. Please let us know if we have to provide more information about that.
>
>
>
> ### **This paper also mentions the works of OOD analysis in programming languages in RELATED WORKS, (a) their relationships with this paper should be discussed in more detail. (b) Why not use their pre-defined scenarios for evaluation?**
>
> a-  Thanks a lot for pointing this out. We clarify the relationship of Bui and Yu [1] with our work in the paper (**in blue**). You can find it in the newly uploaded version.
>
> b- Bui and Yu [1] only focused on the OOD issues in the code classification tasks, which are, in general, easier to define (e.g., removing the data that belongs to a specific label). However, in our work, we are defining OOD scenarios for code generation tasks, which are more widely used in various applications.
>
>
>
> ### **There are many methods proposed in CV and NLP to mitigate catastrophic forgetting in fine-tuning, but only the LoRA method is evaluated, which makes the experimental results not comprehensive enough.**
>
> Thanks a lot for mentioning that. We agree that there are different methods to mitigate catastrophic forgetting. However, in this work, our main focus is to show the performance of two widely used fine-tuning approaches in the source code OOD scenarios. We show that regardless of architectures and parameter sizes, the models' performance dropped in various OOD scenarios (Even for Code Llama 6B, which was fine-tuned using the LoRA method). The paper's main message is to raise awareness about the OOD issues of the fine-tuned LLMs in the source code domain.
>
>
>
> ### **In Section 4.4, what is the performance of the original pre-trained model without FT or LoRA to generate unseen language elements?**
>
> That is an interesting suggestion. We are working on the experiment to provide the results. We are trying to provide the results before the end of the discussion period.
>
>
>
> [1] Nghi D. Q. Bui and Yijun Yu. Energy-bounded learning for robust models of code. CoRR, 2021.

---

> ### Author Response · Authors · 2023-11-21
>
> Here, we provide the results for the following question. We hope that the description and the results resolve your concerns.
> ### **In Section 4.4, what is the performance of the original pre-trained model without FT or LoRA to generate unseen language elements?**
> In the following tables, we present the relative frequencies of generating unseen elements by models fine-tuned using both full fine-tuning (**FT**) and LoRA fine-tuning (**LoRA**) methods (The results from the paper - Figure 3). Furthermore, we included results of generating targeted elements (unseen elements for the fine-tuning cases) using the pre-trained models in zero-shot fashion to demonstrate their performance in generating target elements (**Zero**).
>
>
>
> Table-1 and Table-2 present results for the text-to-code and code refinement tasks, respectively. Table-1 shows that for the zero-shot cases, the generated targeted elements by pre-trained models are on par with LoRA fine-tuning and have higher percentages than the full fine-tuning methods. In general, text-to-code tasks pose challenges, requiring models to map natural language text to targeted codes effectively. Consequently, pre-trained models may find it more difficult to generate desired outputs in this context.
>
> In the zero-shot cases, Table-2 demonstrates that the pre-trained models generate more targeted elements than the fine-tuning methods. This shows that fine-tuning deteriorates previously acquired knowledge. Given the nature of the code refinement task, where the objective is to repair the input programs, models find it relatively easier to produce the programs (e.g., by copying the input tokens).
>
>
>
> Note that, in all of the cases for text-to-code and code refinement tasks, the models' accuracy in the zero-shot cases was **~0.0% of relative performance** (relative to the full fine-tuning results).
>
>
>
> **Table-1:** The frequency ratios of the generated unseen language elements over the frequency in ground truth data for the **text-to-code** task. **Zero** refers to the results of the original pre-trained models. **FT** and **LoRA** refer to the results of full fine-tuning and LoRA fine-tuning methods, respectively.
>
> | Models | Zero | FT | LoRA |
> | ---------- | ---- | -- | ---- |
> | CodeT5 | 10.81% | 3.59% | 14.76% |
> | CodeT5+  | 17.67% | 6.66% | 19.19% |
> | Code Llama | 20.99% | - | 19.56% |
>
>
>
> **Table-2:** The frequency ratios of generated unseen language elements over the frequency in ground truth data for **code refinement** task. **Zero** refers to the results of the original pre-trained models. **FT** and **LoRA** refer to the results of full fine-tuning and LoRA fine-tuning methods, respectively.
>
> | Models | Zero | FT | LoRA |
> | ---------- | ---- | -- | ---- |
> | CodeT5 | 57.20% | 16.10% | 29.29% |
> | CodeT5+  | 66.60% | 23.39% | 35.96% |
> | Code Llama | 70.03% | - | 39.36% |

---

### Author Response · Authors · 2023-11-15
**General response**

We thank all the reviewers for their constructive feedback. We appreciate **Reviewer cCst** recognized the problem we studied is **highly meaningful and significant**, and our work **pioneers the investigation of OOD generalization issues in code LLMs**. The reviewers found the submission **well written** **(Reviewer cCst, Reviewer 9kMw)**, **easy to follow** **(Reviewer cCst)** with proposing **comprehensive and diverse OOD scenarios** **(Reviewer cCst, Reviewer fJQ1)**. Furthermore, they found our work **contributes a systematic approach** **(Reviewer HtnZ, Reviewer fJQ1)** and provides **extensive experiments** **(Reviewer cCst, Reviewer fJQ1)** with **sufficient insights and conclusions** **(Reviewer fJQ1)**. Below, we address the common concerns raised by the reviewers.



### **The findings are predictable and OOD generalization performance has already been studied in various NLP and CV tasks** **( Reviewer HtnZ, Reviewer cCst, Reviewer fJQ1).**

While some of the findings might be intuitive, to the best of our knowledge, this is the first work that studies the OOD behavior of large language models (e.g., Code Llama and CodeT5+) in generation tasks. Kumar et al.[1] only focus on computer vision classification tasks. Furthermore, Luo et al.[2] and Chen et al. [3] focus only on the continual learning domain. None of the works in our Related Work analyze the performance of fine-tuned LLMs in OOD scenarios for code or natural language generation tasks.

Furthermore, for the **first time**, we define OOD scenarios for source code data along three code-related diverse dimensions: length, syntax, and semantic. Our results along each dimension raise awareness about the potential OOD issues of fine-tuned source code models. In the next part, we describe how each of these scenarios can provide real-world insight.

**To conclude** this is the first work investigating the OOD issues of fine-tuned LLMs in generation tasks. We investigate the behavior of two widely used fine-tuning methods, revealing both approaches' challenges in handling OOD scenarios. We also show for the **first time** that LoRA fine-tuning outperforms full fine-tuning in OOD scenarios while it requires less computational resources (We are not aware of any work that studies that).



### **Do the three types of code OOD scenarios accurately represent realistic code OOD data?** **(Reviewer HtnZ, Reviewer cCst)**.

We are just beginning to understand the complexity of OOD-related problems for source code data. In this work, we proposed a systematic approach to approximate various real-world OOD scenarios. We provide more details about these scenarios in the below:

**Length-based scenarios:** Using this OOD scenario, we can study the length-generalization ability of models. For example, whether the models can produce longer code with high quality or how well the models can interpolate over the distribution gaps. The future datasets could also lack programs with certain lengths. Our systematic approach and current results can provide insights into the limitations of the models trained using future datasets. For example, our results show that the models' performance drastically dropped in the extrapolation cases even when we only masked out the smaller programs (e.g., CodeT5's relative performance dropped to 0.0% in the text-to-code task).

**Syntax-based** scenarios can represent scenarios where the model did not see specific elements and APIs. Note that the elements and APIs in a programming language can be updated in the real world. Therefore, the syntax-based scenarios can help us to understand and anticipate the potential shortcomings of the models in the event of program language updates.

**Semantic-based scenarios**:  Given a complex enough programming language, we can implement billions (theoretically infinite) of programs with different **semantics**. It is crucial to see to what extent the model can generate unseen or rare functionality(semantics). However, it is unclear how we can model the semantics of the programs. Therefore, in this work, we approximate the programs' functionalities by embedding them into the continuous space.

[1] Ananya Kumar, Aditi Raghunathan, Robbie Matthew Jones, Tengyu Ma, and Percy Liang. Fine-tuning can distort pretrained features and underperform out-of-distribution. In ICLR, 2022

[2] Yun Luo, Zhen Yang, Fandong Meng, Yafu Li, Jie Zhou, and Yue Zhang. An empirical study of catastrophic forgetting in large language models during continual fine-tuning. arXiv, 2023.

[3] Sanyuan Chen, Yutai Hou, Yiming Cui, Wanxiang Che, Ting Liu, and Xiangzhan Yu. Recall and learn: Fine-tuning deep pretrained language models with less forgetting. In EMNLP, 2020.

---

### Author Response · Authors · 2023-11-22

Dear AC and reviewers,

We hope our response and the new results cover your concerns about our work. Please let us know if we can provide any further clarifications. We appreciate a short reply to let us know your thoughts. We would be delighted if our answers encourage you to raise the rating. Thanks a lot for your time.